# Tumor-promoting function of apoptotic caspases by an amplification loop involving ROS, macrophages and JNK in *Drosophila*

Ernesto Pérez[†‡], Jillian L Lindblad[†], Andreas Bergmann*

Department of Molecular, Cell and Cancer Biology, University of Massachusetts Medical School, Worcester, United States

**Abstract** Apoptosis and its molecular mediators, the caspases, have long been regarded as tumor suppressors and one hallmark of cancer is 'Evading Apoptosis'. However, recent work has suggested that apoptotic caspases can also promote proliferation and tumor growth under certain conditions. How caspases promote proliferation and how cells are protected from the potentially harmful action of apoptotic caspases is largely unknown. Here, we show that although caspases are activated in a well-studied neoplastic tumor model in *Drosophila*, oncogenic mutations of the proto-oncogene Ras (*Ras^{V12}*) maintain tumorous cells in an 'undead'-like condition and transform caspases from tumor suppressors into tumor promotors. Instead of killing cells, caspases now promote the generation of intra- and extracellular reactive oxygen species (ROS). One function of the ROS is the recruitment and activation of macrophage-like immune cells which in turn signal back to tumorous epithelial cells to activate oncogenic JNK signaling. JNK further promotes and amplifies caspase activity, thereby constituting a feedback amplification loop. Interfering with the amplification loop strongly reduces the neoplastic behavior of these cells and significantly improves organismal survival. In conclusion, *Ras^{V12}*-modified caspases initiate a feedback amplification loop involving tumorous epithelial cells and macrophage-like immune cells that is necessary for uncontrolled tumor growth and invasive behavior.

DOI: https://doi.org/10.7554/eLife.26747.001

*For correspondence: andreas.bergmann@umassmed.edu

[†]These authors contributed equally to this work

**Present address:** [‡]Department of Biology, Austin College, Sherman, United States

**Competing interests:** The authors declare that no competing interests exist.

## Introduction

Larval imaginal discs in *Drosophila* are single-cell layered sacs of epithelial cells that develop into the adult appendages such as eyes and wings, and are frequently used as genetic models for growth control and tumor development. Maintenance of apical-basal polarity of epithelial cells is critical for suppression of neoplastic tumor development (*Elsum et al., 2012*; *Bergstralh and St Johnston, 2012*; *Martin-Belmonte and Perez-Moreno, 2011*). Mutations in genes encoding components of the Scribble complex including *scribble* (*scrib*), *lethal giant larvae* (*lgl*) and *discs large* (*dlg*) disrupt apical-basal polarity in epithelial cells which can trigger malignant neoplastic tumor growth (*Bergstralh and St Johnston, 2012*; *Elsum et al., 2012*; *Bilder et al., 2000*; *Gateff, 1978*; *Bilder and Perrimon, 2000*). *Drosophila* larvae entirely mutant for *scrib* fail to respond to stop signals of growth, fail to pupariate and continue to grow as larvae (*Gateff, 1994*; *Wodarz, 2000*). They die as giant larvae with severely overgrown imaginal discs.

However, *scrib* mutant cells (clones) in otherwise wild-type imaginal discs are eliminated by cell competition mediated by neighboring wild-type cells (*Brumby and Richardson, 2003*; *Menéndez et al., 2010*; *Igaki et al., 2009*; *Uhlirova et al., 2005*; *Ohsawa et al., 2011*; *Leong et al., 2009*; *Chen et al., 2012*; *Vaughen and Igaki, 2016*). Mechanistically, in response to

**eLife digest** Throughout the development and life of an animal, many of its cells die and are removed to make space for new tissues. A group of proteins called caspases play a key role in this cell death process. Chemotherapy and radiotherapy work as cancer treatments because they damage dividing cancer cells to such an extent that caspases are turned on and kill those cells. Unfortunately the tumor occasionally grows back after the treatment, partly because caspases also help to produce signals that boost the growth of any surviving cancer cells.

Pérez, Lindblad and Bergmann have now used the fruit fly *Drosophila* to investigate how caspases encourage tumor cells to grow and spread. The fruit flies all carried a mutation in a gene called *Ras*, which is often mutated in human cancers. Using a combination of genetic and biochemical experiments Pérez, Lindblad and Bergmann found that mutant *Ras* prevents cells with active caspases from dying. Instead, the caspases redirect their activity and help cancer cells to produce small chemicals called reactive oxygen species. These chemicals can play many different roles in cancers, but in this setting they attract immune cells to the site of the tumor. The immune cells in turn send other signals back to the cancer cells, which further activate the caspases. Overall, this self-perpetuating signaling loop between the cancer cells and the surrounding immune cells helps the tumors to grow.

Future work toward developing new cancer treatments will need to work on ways of enhancing the cell-killing properties of caspases while inhibiting their ability to help tumors to grow. Further experiments will also be needed to find out exactly how the mutant *Ras* gene protects tumor cells from death.

DOI: https://doi.org/10.7554/eLife.26747.002

cell competition, Eiger, the Tumor Necrosis Factor alpha (TNFα)-like ligand in *Drosophila*, triggers Jun N-terminal kinase (JNK) activation and apoptosis in *scrib* mutant cells (**Igaki et al., 2009**; **Brumby and Richardson, 2003**; **Uhlirova et al., 2005**; **Cordero et al., 2010**; **Ohsawa et al., 2011**; **Leong et al., 2009**; **Igaki et al., 2006**; **Chen et al., 2012**). This tumor-suppressing function is dependent on Eiger and JNK through induction of apoptosis. Inhibition of Eiger or JNK restores the growth potential of *scrib* mutant cells which can then form large tumor masses in imaginal discs (**Brumby and Richardson, 2003**; **Igaki et al., 2009**; **Uhlirova et al., 2005**; **Chen et al., 2012**).

However, if additional oncogenic mutations such as $Ras^{V12}$ are introduced into *scrib* mutant cells (referred to a $scrib^{-/-}\ Ras^{V12}$), they can unleash their full malignant potential (**Brumby and Richardson, 2003**; **Pagliarini and Xu, 2003**). $scrib^{-/-}\ Ras^{V12}$ mosaic eye/antennal imaginal discs display all neoplastic features observed in human tumors including unrestricted growth, failure to differentiate, tissue invasion and organismal lethality (**Pagliarini and Xu, 2003**; **Brumby and Richardson, 2003**). $scrib^{-/-}\ Ras^{V12}$ clones occupy a large portion of the mosaic disc and trigger multi-layered overgrowth of the entire disc compared to wild-type controls (**Figure 1H,I**). $scrib^{-/-}\ Ras^{V12}$ mutant cells also invade other tissues, most notably the ventral nerve cord (VNC) in the brain (**Figure 1H,I**) (**Pagliarini and Xu, 2003**). The $scrib^{-/-}\ Ras^{V12}$ condition in *ey-FLP*-induced eye imaginal disc mosaics is 100% lethal. 95% of *ey-FLP*-induced $scrib^{-/-}\ Ras^{V12}$ mosaic animals die as larvae; the remaining animals die during pupal stages.

Interestingly, $Ras^{V12}$ inhibits the apoptotic activity of JNK and converts the tumor-suppressor function of Eiger and JNK in $scrib^{-/-}$ cells into a tumor-promoting one in $scrib^{-/-}\ Ras^{V12}$ cells (**Enomoto et al., 2015**; **Cordero et al., 2010**; **Uhlirova et al., 2005**; **Igaki et al., 2006**; **Uhlirova and Bohmann, 2006**). Therefore, the aggressive tumor growth of $scrib^{-/-}\ Ras^{V12}$ mutant clones becomes dependent on Eiger and JNK (**Igaki et al., 2006**; **Uhlirova and Bohmann, 2006**; **Brumby et al., 2011**). Mechanistically, it is not understood how $Ras^{V12}$ promotes this oncogenic switch of Eiger and JNK.

Caspases are Cys-proteases that mediate the mechanistic events of apoptotic cell death (**Shalini et al., 2015**; **Xu et al., 2009**; **Fuchs and Steller, 2011**; **Salvesen et al., 2016**). They are synthesized as zymogens and depending on the length of their prodomains can be classified into initiator and effector caspases. Initiator caspases such as mammalian Caspase-9 or its *Drosophila* ortholog Dronc are controlled by upstream signaling events and when activated initiate apoptosis

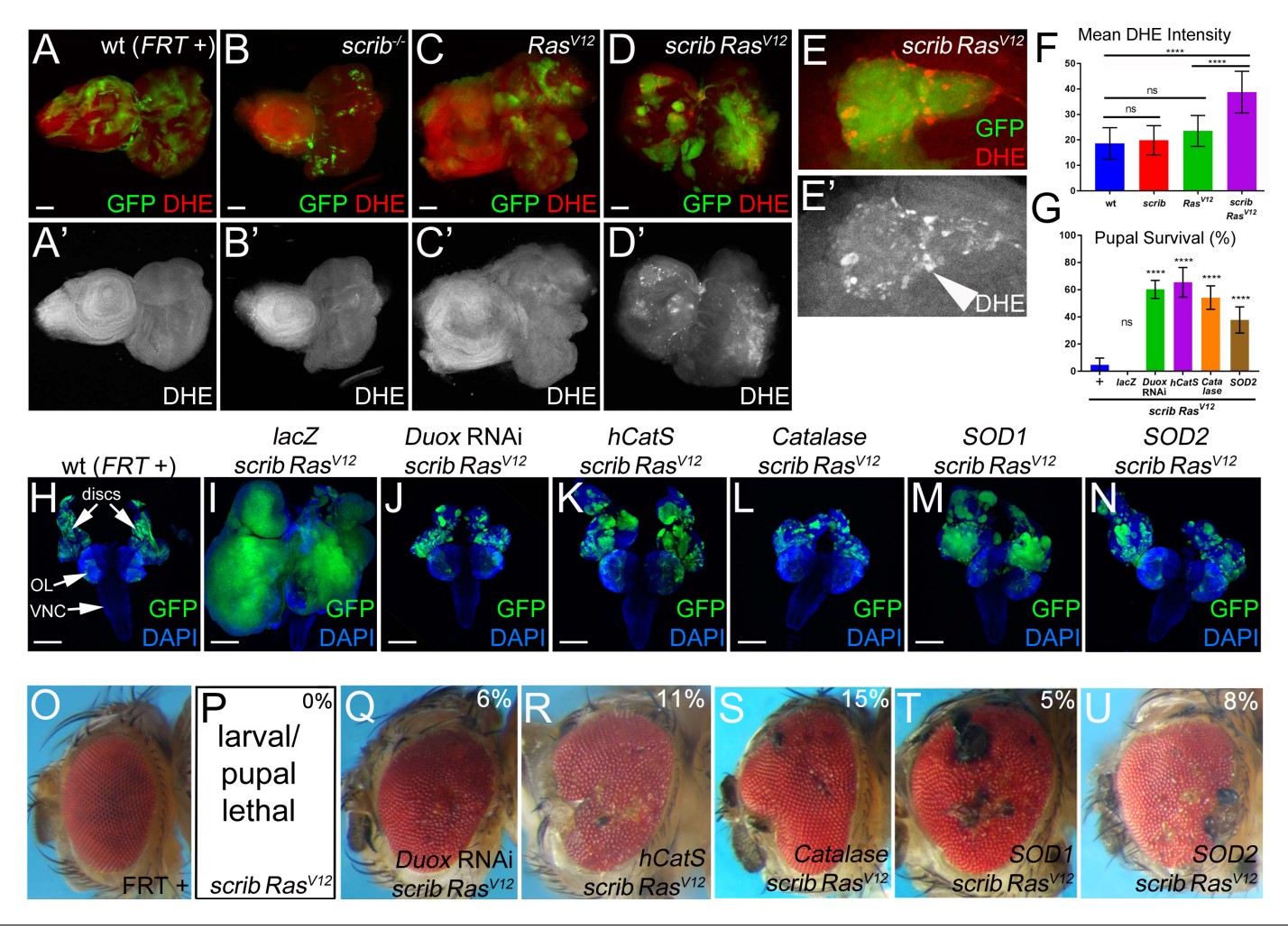

**Figure 1.** Both intra- and extracellular ROS contribute to the strong neoplastic phenotype of *scrib*$^{-/-}$ *Ras*$^{V12}$. Mosaic animals in this and subsequent figures were obtained using the MARCM technique (*Lee and Luo, 1999*) with *ey-FLP* (*Newsome et al., 2000*) to induce mitotic recombination in eye imaginal discs. GFP depicts MARCM clones. Posterior is to the right. (A–D') Wild-type (wt, *FRT +*) (A), *scribble* (*scrib*$^{-/-}$) (B), *Ras*$^{V12}$–expressing (C) and *scrib*$^{-/-}$ *Ras*$^{V12}$ (D) eye/antennal mosaic imaginal discs from third instar larvae labeled with the ROS indicator Dihydroethidium (DHE). Scale bars: 50 µm. (E) Enlarged *scrib*$^{-/-}$ *Ras*$^{V12}$ clones labeled for DHE. Arrowhead in (E') marks a cell of high DHE labeling. (F) DHE quantification reveals that ROS levels are significantly higher in *scrib*$^{-/-}$ *Ras*$^{V12}$ mutant clones compared to wt (*FRT +*), *scrib*$^{-/-}$ or *Ras*$^{V12}$-expressing clones. Plotted is the mean signal intensity ±SD of DHE labelings in clones, analyzed by one-way ANOVA with Holm-Sidak test for multiple comparisons. ****p<0.0001; ns – not significant. Multiple clones from five to ten discs of each genotype were analyzed. (G) Reduction of extra- and intracellular ROS levels in *scrib*$^{-/-}$ *Ras*$^{V12}$ mutant clones significantly improves the pupariation rates of animals bearing *scrib Ras*$^{V12}$ mosaic eye imaginal discs. Expression of *UAS-lacZ* in *scrib*$^{-/-}$ *Ras*$^{V12}$ clones as control has no effect on the pupariation rate. Pupariation rates were determined as the ratio of late stage mutant pupae vs total pupae and were analyzed by one-way ANOVA with Holm-Sidak test for multiple comparisons. Error bars are SD. P values are relative to *scrib*$^{-/-}$ *Ras*$^{V12}$ results (left column) and are indicated above the experimental columns. ****p<0.0001; ns – not significant. At least 100 pupae were counted per genotype. Experiments were performed three times. (H–N) Cephalic complexes composed of eye/antennal discs, optic lobes (OL) and ventral nerve cord (VNC) from day 11 old third instar larvae. The genotype is indicated on top of each panel. Expression of *UAS-lacZ* served as negative control (I). Depletion of ROS strongly reduces clone size (green) and normalizes growth in (J–N). DAPI (blue) labels the outline of the tissue. Scale bars: 200 µm. (O–U) Adult eyes of control (O) and *scrib*$^{-/-}$ *Ras*$^{V12}$ mosaics expressing the indicated antioxidant transgenes (Q–U). The percentage number in the top right of each panel indicates the adult survival rate relative to pupal survival. Note that *ey-FLP*-induced *scrib*$^{-/-}$ *Ras*$^{V12}$ MARCM mosaics are 100% lethal (0% adult survival) (P). Genotypes: (A,H,O) *yw ey-FLP/+; act>y$^+$>Gal4, UAS-GFP$^{56ST}$/+; FRT82B tub-Gal80/. FRT82B w$^+$*; (B) *yw ey-FLP/+; act>y$^+$>Gal4, UAS-GFP$^{56ST}$/+; FRT82B tub-Gal80/. FRT82B scrib$^2$*; (C) *yw ey-FLP/+; act>y$^+$>Gal4, UAS-GFP$^{56ST}$/UAS-Ras$^{V12}$; FRT82B tub-Gal80/FRT82B w$^+$*; (D,E,P) *yw ey-FLP/+; act>y$^+$>Gal4, UAS-GFP$^{56ST}$/+; FRT82B tub-Gal80/UAS-Ras$^{V12}$ FRT82B scrib$^2$*; (I–N,Q–U) *yw ey-FLP/+; act>y$^+$>Gal4, UAS-GFP$^{56ST}$/UAS-X; FRT82B tub-Gal80/UAS-Ras$^{V12}$ FRT82B scrib$^2$* with *UAS-X* being *UAS-lacZ* (I), *UAS-Duox*$^{RNAi}$ (J,Q), *UAS-hCatS* (K,R), *UAS-Catalase* (L,S), *UAS-SOD1* (M,T) and *UAS-SOD2* (N,U).

DOI: https://doi.org/10.7554/eLife.26747.003

*Figure 1 continued on next page*

*Figure 1 continued*

The following figure supplements are available for figure 1:

**Figure supplement 1.** Caspase-dependent generation of ROS as revealed by H$_2$DCF-DA labelings.
DOI: https://doi.org/10.7554/eLife.26747.004

**Figure supplement 2.** Strong induction of β-Gal by expression of *UAS-lacZ* in *scrib*$^{-/-}$ *Ras*$^{V12}$ clones.
DOI: https://doi.org/10.7554/eLife.26747.005

by activating effector caspases such as mammalian Caspase-3 or its *Drosophila* ortholog DrICE (*Fuchs and Steller, 2011*; *Shalini et al., 2015*; *Salvesen et al., 2016*). Caspases induce apoptosis of many cells to maintain homeostatic conditions, and are also thought to be critical for tumor suppression by eliminating malignant cells.

However, caspases can also have tumor-promoting roles, for example through apoptosis-induced proliferation (AiP), a caspase-driven process by which apoptotic cells produce mitogenic signals for proliferation of neighboring surviving cells (*Mollereau et al., 2013*) (reviewed in [*Fogarty and Bergmann, 2017*; *Ryoo and Bergmann, 2012*]). There are two types of AiP. During 'genuine' AiP, apoptotic cells release mitogenic factors before completing the apoptotic program. This type of AiP has been described for regeneration and wound healing both in vertebrates and invertebrates (*Tseng et al., 2007*; *Fan and Bergmann, 2008*; *Chera et al., 2009*; *Li et al., 2010*). 'Genuine' AiP may also be involved in human pathologies such as cancer, and may account for increased cell proliferation and repopulation of tumors following cytotoxic treatments (chemo- or radiotherapy) which induces massive apoptosis (reviewed in [*Fogarty and Bergmann, 2017*; *Ichim and Tait, 2016*]). Caspases play significant tumor-promoting roles in these settings (*Li et al., 2010*; *Huang et al., 2011*; *Donato et al., 2014*; *Cheng et al., 2015*; *Zhang et al., 2015*; *Kurtova et al., 2015*).

The second type is 'undead' AiP. Here, the apoptosis pathway is induced upstream, but the execution of apoptosis is blocked. In *Drosophila*, apoptosis inhibition is achieved experimentally by expression of the effector caspase inhibitor p35 which very specifically inhibits DrICE (*Hay et al., 1994*; *Meier et al., 2000*; *Hawkins et al., 2000*). Therefore, because these cells have initiated the apoptotic process and contain active Dronc, but cannot die, they are referred to as 'undead'. In 'undead' cells, non-apoptotic functions of active Dronc can now be examined, one of which is the release of mitogenic signals for induction of AiP which can lead to hyperplastic overgrowth (*Wells et al., 2006*; *Kondo et al., 2006*; *Huh et al., 2004*; *Ryoo et al., 2004*; *Martín et al., 2009*; *Pérez-Garijo et al., 2009*; *Pérez-Garijo et al., 2004*; *Fan et al., 2014*; *Rudrapatna et al., 2013*; *Pérez-Garijo et al., 2005*). 'Undead' states of cells may also be present under pathological conditions. For instance, it has been proposed that *Ras*$^{V12}$ can maintain apoptotic cells in an 'undead'-like state promoting tumorigenesis (*Hirabayashi et al., 2013*).

Mechanistically, we have shown that AiP-mediated hyperplastic overgrowth of 'undead' tissue depends on a Dronc-initiated feedback amplification loop which involves reactive oxygen species (ROS) – specifically extracellular ROS produced by the membrane-bound NADPH oxidase Duox –, activation of macrophage-like hemocytes, secretion of Eiger by hemocytes, Eiger-dependent activation of JNK in epithelial disc and further activation of Dronc by JNK (*Fogarty et al., 2016*) (reviewed by [*Diwanji and Bergmann, 2017a*; *Diwanji and Bergmann, 2017b*]). Therefore, similar to the *scrib*$^{-/-}$ *Ras*$^{V12}$ case, Eiger and JNK signaling have proliferation- and growth-promoting functions in this 'undead' AiP model.

These similarities prompted us to investigate the role of ROS and caspases for tumor growth of *scrib*$^{-/-}$ *Ras*$^{V12}$ clones in *Drosophila*. We report that oncogenic Ras switches the pro-apoptotic activity of caspases into a tumor-promoting one and thereby maintains *scrib*$^{-/-}$ *Ras*$^{V12}$ cells in an 'undead'-like state. Consistently, inhibition of caspases blocks tumor growth and tissue invasion. The tumor-promoting function of apoptotic caspases is dependent on the generation of intra- and extracellular ROS which are required for neoplastic behavior of *scrib*$^{-/-}$ *Ras*$^{V12}$ clones. Furthermore, caspase-induced ROS are essential for the recruitment and activation of hemocytes at *scrib*$^{-/-}$ *Ras*$^{V12}$ mosaic discs. Hemocytes signal back to tumorous epithelial cells to stimulate JNK signaling which further promotes caspase activity. Thus, these events constitute a feedback amplification loop which is necessary for neoplastic activity of *scrib*$^{-/-}$ *Ras*$^{V12}$ cells. This work extends previous models about the conversion of Eiger and JNK signaling from anti-tumor to pro-tumor roles by oncogenic Ras and

identifies caspases as essential components of this switch. In conclusion, although apoptotic caspases are usually considered to be tumor suppressors, under certain conditions, for example in the presence of oncogenic $Ras^{V12}$ in $scrib$ mutant cells, they can also adopt a tumor-promoting role.

## Results

### ROS are required for neoplastic characteristics of $scrib^{-/-}$ $Ras^{V12}$ mosaic discs

Recently, in a model of 'undead' AiP, we showed that Duox-generated ROS are important for activation of hemocytes, JNK signaling and hyperplastic overgrowth (*Fogarty et al., 2016*; *Diwanji and Bergmann, 2017a*; *Diwanji and Bergmann, 2017b*). Therefore, we examined if ROS have a similar function in the neoplastic $scrib^{-/-}$ $Ras^{V12}$ tumor model in *Drosophila*. $scrib^{-/-}$ $Ras^{V12}$ clones were induced by MARCM using *ey-FLP* in eye/antennal imaginal discs, the traditional tissue for this model (*Brumby and Richardson, 2003*; *Pagliarini and Xu, 2003*). The ROS indicator dihydroethidium (DHE) strongly labels $scrib^{-/-}$ $Ras^{V12}$ mutant clones in mosaic discs, while wild-type (wt), $scrib^{-/-}$ and $Ras^{V12}$-expressing clones are not labeled by DHE or very little (*Figure 1A–D'*; quantified in *Figure 1F*; see also *Figure 2A,A'*). Similar results were reported recently (*Katheder et al., 2017*; *Manent et al., 2017*). A different ROS indicator, $H_2$DCF-DA, confirms these results (*Figure 1—figure supplement 1A–C,G*). $scrib^{-/-}$ $Ras^{V12}$ clones display an increased diffuse cytosolic DHE labeling (*Figure 1E,E'*). At the boundary of $scrib^{-/-}$ $Ras^{V12}$ clones, several cells stain very intensely for DHE (*Figure 1E,E'*; arrow head).

To examine the function of ROS in this neoplastic tumor model, we reduced their amount either by down-regulating ROS-producing enzymes such as Duox or by overexpressing ROS-removing enzymes such as catalases and superoxide dismutases (SOD). As shown previously, $scrib^{-/-}$ $Ras^{V12}$ occupy a large portion of the eye/antennal imaginal disc and display a strong neoplastic tumor phenotype (*Pagliarini and Xu, 2003*) (see also *Figure 2F*). Because expression of the antioxidant enzymes was achieved by the UAS/Gal4 system, we tested first whether increasing the *UAS* gene dose may modify (suppress) the tumor phenotype and expressed an unrelated control gene, *UAS-lacZ*, in $scrib^{-/-}$ $Ras^{V12}$ clones. However, despite strong expression of β-Gal (*Figure 1—figure supplement 2*), the tumor phenotype of $scrib^{-/-}$ $Ras^{V12}$ eye/antennal imaginal discs is not significantly suppressed by addition of an additional *UAS*-driven transgene (*Figure 1I*).

In contrast, removing extracellular ROS by *UAS-Duox* RNAi or overexpression of the *UAS-hCatS* transgene which encodes a secreted human catalase (*Ha et al., 2005b2005*; *Ha et al., 2005a*), strongly suppressed tumor growth of $scrib^{-/-}$ $Ras^{V12}$ mutant cells (*Figure 1J,K*) suggesting that extracellular ROS are required for tumor growth. Interestingly and in contrast to the 'undead' AiP model, removing intracellular ROS by misexpression of intracellular *Catalase*, *SOD1* and *SOD2* also strongly suppressed tumor growth (*Figure 1L–N*). These results suggest that both intra- and extracellular ROS are required for tumor growth of $scrib^{-/-}$ $Ras^{V12}$ clones.

Importantly also, reduction of ROS strongly reduces the invasive behavior of $scrib^{-/-}$ $Ras^{V12}$ mutant cells (*Figure 1J–N*) which significantly improves the survival rate of the affected animals. Compared to *ey* >MARCM $scrib^{-/-}$ $Ras^{V12}$ mutant larvae, of which only 5% reach pupal stages, between 40% and 70% of the *ey* >MARCM $scrib^{-/-}$ $Ras^{V12}$ larvae expressing antioxidant enzymes develop into pupae (*Figure 1G*). Expression of the *UAS-lacZ* control transgene does not improve pupal survival (*Figure 1G*). Furthermore, we also recovered viable adult *ey* >MARCM $scrib^{-/-}$ $Ras^{V12}$ mosaic animals expressing antioxidant genes, although at a low rate (5–15% of the surviving pupae), which was never observed for $scrib^{-/-}$ $Ras^{V12}$ only (*Figure 1P–U*). Although their eyes and heads are deformed and weakly overgrown often with necrotic patches compared to wt control (*Figure 1O,Q–U*), these animals live! Previously, only few examples of surviving $scrib^{-/-}$ $Ras^{V12}$ mosaic animals have been reported. In these examples, viable $scrib^{-/-}$ $Ras^{V12}$ mosaic animals were recovered when essential steps in tumor development such as oncogenic JNK signaling or the cell cycle were inhibited in $scrib^{-/-}$ $Ras^{V12}$ clones (*Brumby and Richardson, 2003*; *Külshammer et al., 2015*; *Külshammer and Uhlirova, 2013*). Therefore, the observation that reduction of ROS suppresses tumor growth and enhances organismal survival strongly suggests that ROS play a very significant role for the neoplastic characteristics of $scrib^{-/-}$ $Ras^{V12}$ animals.

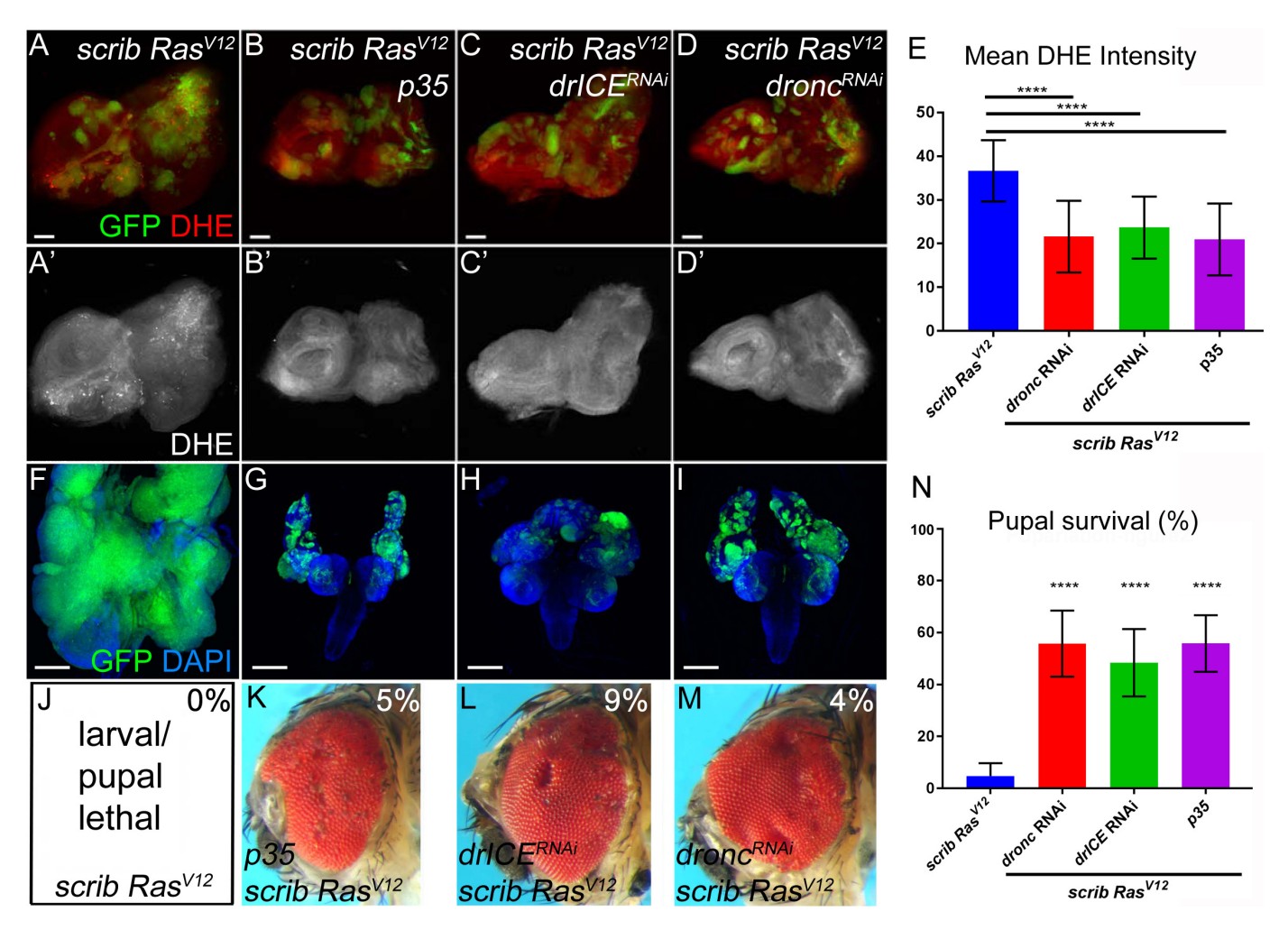

**Figure 2.** Caspases are required for ROS generation and neoplastic overgrowth in $scrib^{-/-}$ $Ras^{V12}$. (A–D') Expression of the effector caspase inhibitor $p35$ (B), $drICE$ RNAi (C) and $dronc$ RNAi (D) suppresses $scrib^{-/-}$ $Ras^{V12}$ clone size (green) and ROS generation in $scrib^{-/-}$ $Ras^{V12}$ clones. The (') panels indicate the labeling of the ROS indicator DHE (grey). Scale bars: 50 μm. (E) DHE quantification reveals that ROS levels are significantly reduced in $scrib^{-/-}$ $Ras^{V12}$ mutant clones with reduced or inhibited caspase activity. Shown is the mean signal intensity ±SD of DHE labelings in clones, analyzed by one-way ANOVA with Holm-Sidak test for multiple comparisons. ****p<0.0001. Multiple clones from five to ten discs of each genotype were analyzed. (F–I) The growth and invasion of cephalic complexes of 11 day old $scrib^{-/-}$ $Ras^{V12}$ larvae (F) is strongly suppressed by p35 (G), $drICE$ RNAi (H) and $dronc$ RNAi (I). Clone size (green) in (F–I) is strongly reduced. DAPI labels the outline of the tissue. Scale bars: 200 μm. (J–M) Adult eyes of surviving $scrib^{-/-}$ $Ras^{V12}$ animals expressing $p35$ (K), $drICE$ RNAi (L) and $dronc$ RNAi (M). The percentage number in the top right of each panel indicates the adult survival rate relative to pupal survival. (N) Reduction or inhibition of caspase activity in $scrib^{-/-}$ $Ras^{V12}$ mutant clones significantly improves the pupariation rates of animals bearing $scrib^{-/-}$ $Ras^{V12}$ mosaic eye imaginal discs. Pupariation rates were determined as the ratio of late stage mutant pupae vs total pupae and were analyzed by one-way ANOVA with Holm-Sidak test for multiple comparisons. Error bars are SD. P values are relative to $scrib^{-/-}$ $Ras^{V12}$ results (left column) and are indicated above the experimental columns. ****p<0.0001. At least 100 pupae were counted per genotype. Experiments were performed three times. **Genotypes**: (A,F,J) $yw$ $ey$-FLP/+; $act$>$y^{+}$>Gal4, UAS-GFP$^{56ST}$/+; FRT82B tub-Gal80/UAS-Ras$^{V12}$ FRT82B scrib$^{2}$; (B–D,G–I,K–M) $yw$ $ey$-FLP/+; $act$>$y^{+}$>Gal4, UAS-GFP$^{56ST}$/UAS-X; FRT82B tub-Gal80/UAS-Ras$^{V12}$ FRT82B scrib$^{2}$ with UAS-X being UAS-p35 (B,G,K), UAS-drICE$^{RNAi}$ (C,H,L) and UAS-dronc$^{RNAi}$ (D,I,M).
DOI: https://doi.org/10.7554/eLife.26747.006

## Caspases promote tumors by inducing the generation of ROS

In 'undead' cells, the initiator caspase Dronc (caspase-9 ortholog) has been shown to stimulate the production of ROS (*Fogarty et al., 2016*). Therefore, we examined the role of Dronc as well as the effector caspase DrICE (caspase-3 ortholog) for generation of ROS and tumorous overgrowth in $scrib^{-/-}$ $Ras^{V12}$ mosaic eye discs. As an additional assay, we expressed the effector caspase inhibitor

p35 in $scrib^{-/-}$ $Ras^{V12}$ mutant clones. Removing or inhibiting caspases in $scrib^{-/-}$ $Ras^{V12}$ mutant clones strongly reduced DHE labeling suggesting suppression of ROS generation (*Figure 2A–D'*; quantified in *Figure 2E*). Similar results were obtained with a different ROS indicator, $H_2$DCF-DA (*Figure 1—figure supplement 1C–G*). Consequently, tumor overgrowth and invasion of the VNC is dramatically reduced upon removal or inhibition of caspases in $scrib^{-/-}$ $Ras^{V12}$ mutant cells (*Figure 2F–I*). Reduction of caspase activity also increases pupal survival (*Figure 2N*) and viable animals with mosaic $scrib^{-/-}$ $Ras^{V12}$ heads and eyes were recovered as adults at a rate of 4–9% of the surviving pupae (*Figure 2J–M*).

The requirement of caspases for generation of ROS and neoplastic behavior suggests that caspases are activated in $scrib^{-/-}$ $Ras^{V12}$ mutant cells. To verify this, we labeled $scrib^{-/-}$ $Ras^{V12}$ mosaic eye discs with cleaved caspase-3 (CC3) antibody which detects activated (cleaved) effector caspases and an unknown non-apoptotic substrate of Dronc (*Fan and Bergmann, 2010*; *Srinivasan et al., 1998*). Indeed, while there is very little CC3 labeling in mosaic control discs (FRT +), $scrib^{-/-}$ $Ras^{V12}$ mutant clones label significantly stronger with CC3 antibody (*Figure 3A–B'*, yellow arrowhead; quantified in *Figure 3D*). In addition to the CC3 labeling in $scrib^{-/-}$ $Ras^{V12}$ clones, there is also staining immediately outside the clones which appears even more intense than the labeling inside the clones (*Figure 3B,B'*; white arrows). In fact, quantification reveals that this non-autonomous CC3 labeling is 2 to 2.5-fold higher than autonomous CC3 labeling in $scrib^{-/-}$ $Ras^{V12}$ clones (*Figure 3D*). Both, CC3 labeling inside and outside of $scrib^{-/-}$ $Ras^{V12}$ clones are autonomously dependent on the caspases DrICE (*Figure 3C*) and Dronc as well as on ROS (*Figure 3—figure supplement 1*).

Labeling with the CC3 antibody indicates active Dronc both in apoptotic and non-apoptotic cells (*Fan and Bergmann, 2010*; *Fan et al., 2014*). Therefore, because of the strong tumor growth of $scrib^{-/-}$ $Ras^{V12}$ mosaics, we wondered if the CC3-positive cells in $scrib^{-/-}$ $Ras^{V12}$ clones are actually apoptotic and labeled $scrib^{-/-}$ $Ras^{V12}$ discs with TUNEL, an apoptotic assay that detects DNA fragmentation, a hallmark of apoptosis downstream of effector caspases (*Gavrieli et al., 1992*). Interestingly, although a few TUNEL-positive cells are detectable within $scrib^{-/-}$ $Ras^{V12}$ clones, the majority of TUNEL-positive cells are located outside the clones (*Figure 3E,E'*; arrows). Quantification reveals that almost 90% of all apoptotic cells in $scrib^{-/-}$ $Ras^{V12}$ mosaic discs are outside the mutant clones (*Figure 3F*).

These observations allow us to make an important conclusion. Despite detectable caspase (Dronc) activity in $scrib^{-/-}$ $Ras^{V12}$ mutant clones by CC3 labeling, this activity does not appear to trigger a significant amount of apoptosis in these clones. In contrast, our genetic analysis suggest that the strong tumor growth phenotype of $scrib^{-/-}$ $Ras^{V12}$ mosaic eye discs is dependent on caspases (*Figure 2*) suggesting that they have adopted a tumor-promoting function. This is surprising as caspase activity and apoptosis in general are thought to act as tumor suppressors (*Hanahan and Weinberg, 2011*; *Hanahan and Weinberg, 2000*). In fact, caspase activity in $scrib^{-/-}$ single mutant cells does act as a tumor suppressor by killing them (*Brumby and Richardson, 2003*; *Igaki et al., 2006*; *Igaki et al., 2009*; *Uhlirova et al., 2005*; *Chen et al., 2012*). In contrast, in $scrib^{-/-}$ $Ras^{V12}$ mutant cells, this caspase activity persists, but does not appear to induce a significant amount of apoptosis. Therefore, these data suggest that $Ras^{V12}$ maintains $scrib^{-/-}$ cells in an 'undead'-like condition, consistent with a previous report (*Hirabayashi et al., 2013*). Furthermore, $Ras^{V12}$ changes the activity of caspases to adopt a tumor-promoting role.

## Hemocyte recruitment to $scrib^{-/-}$ $Ras^{V12}$ tumors depends on caspase-generated ROS

Next, we examined the role of ROS for neoplastic growth of $scrib^{-/-}$ $Ras^{V12}$ mosaic discs. One known function of extracellular ROS is the recruitment and activation of *Drosophila* macrophages (hemocytes) in the 'undead' AiP model (*Diwanji and Bergmann, 2017a*; *Fogarty et al., 2016*; *Diwanji and Bergmann, 2017b*). Hemocytes have been shown to be associated with $scrib^{-/-}$ $Ras^{V12}$ mosaic discs (*Cordero et al., 2010*; *Pastor-Pareja et al., 2008*; *Külshammer and Uhlirova, 2013*). Therefore, we tested if ROS contribute to the recruitment and activation of hemocytes to $scrib^{-/-}$ $Ras^{V12}$ mosaic eye imaginal discs. At control discs, hemocytes adhere in small cellular aggregates, mostly at the antennal disc (*Figure 4A,A',L*). In contrast, they are recruited to neoplastic $scrib^{-/-}$ $Ras^{V12}$ tumor sites in large numbers where they cover the eye portion of the disc (*Figure 4B,B'*; quantified in *Figure 4K*) consistent with previous reports (*Cordero et al., 2010*; *Pastor-Pareja et al., 2008*; *Külshammer and Uhlirova, 2013*). They also change their morphological

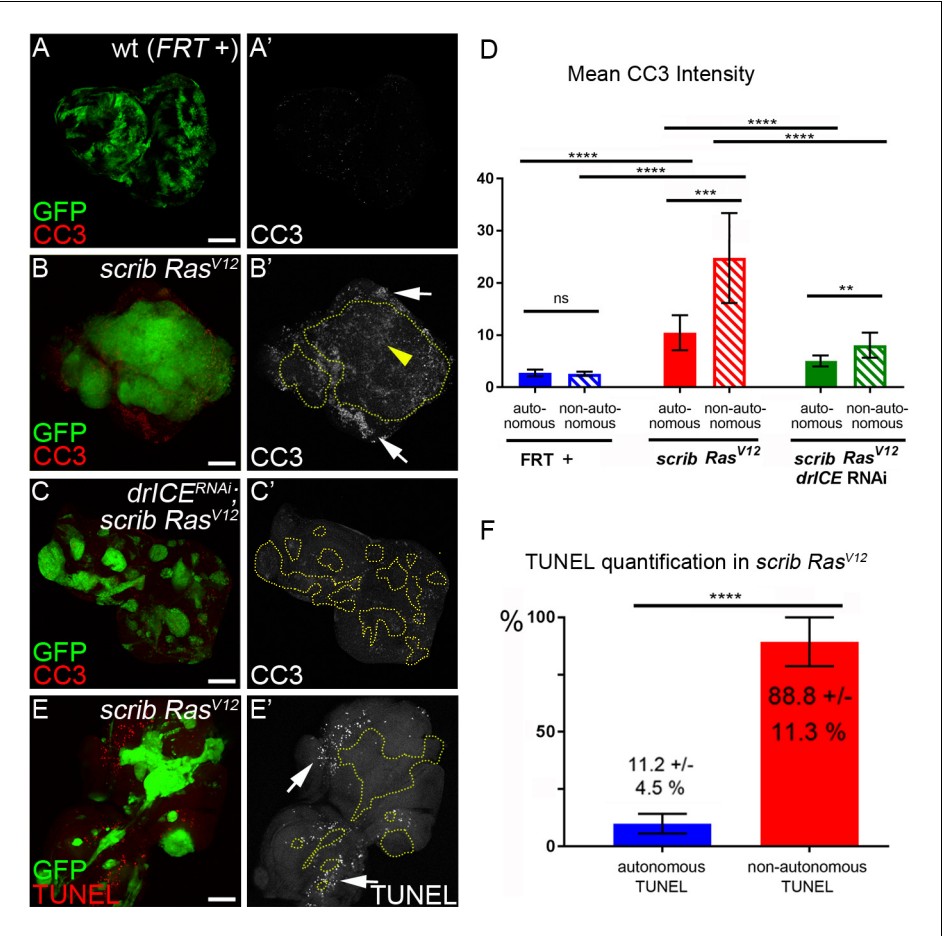

**Figure 3.** Analysis of caspase activity and apoptosis in scrib⁻/⁻ Ras^V12 mosaic eye discs. (A–C') Cleaved caspase 3 (CC3) (red in (A–C); grey in (A'–C')) analysis of control (wt, FRT +) (A), scrib⁻/⁻ Ras^V12 (B) and scrib⁻/⁻ Ras^V12 expressing drICE RNAi (C) mosaic eye imaginal discs. CC3 labeling is detectable autonomously (yellow arrowhead) and non-autonomously (white arrows) of scrib⁻/⁻ Ras^V12 mutant clones (B'). Clones in (B') and (C') are outlined by yellow, dotted lines. Scale bars: 50 μm. (D) CC3 quantification of mosaic FRT + (control), scrib⁻/⁻ Ras^V12 and drICE^RNAi;scrib⁻/⁻ Ras^V12 eye-antennal imaginal discs reveals significant increase of caspase activity both inside (autonomously) and outside (non-autonomously) of scrib⁻/⁻ Ras^V12 clones. Plotted is the mean signal intensity ±SD of autonomous and non-autonomous CC3 labelings, immediately adjacent to the clones. Analysis was performed by one-way ANOVA with Holm-Sidak test for multiple comparisons. ****p<0.0001; **p<0.01; ns – not significant. Ten discs per genotype were analyzed. (E) TUNEL assay as an apoptotic marker of scrib⁻/⁻ Ras^V12 mosaic eye discs. White arrows mark TUNEL-positive cells outside scrib⁻/⁻ Ras^V12 clones, outlined by yellow dotted lines. (F) TUNEL quantification reveals that almost 90% of apoptotic cells in scrib⁻/⁻ Ras^V12 mosaic discs are outside of mutant clones. Autonomous and non-autonomous counts of TUNEL-positive cells were analyzed by paired student's t-test. ****p<0.0001. The distribution of TUNEL-positive cells in seven discs is plotted. **Genotypes**: (A) yw ey-FLP/+; act>y⁺>Gal4, UAS-GFP^56ST/+; FRT82B tub-Gal80/. FRT82B w⁺; (B,E) yw ey-FLP/+; act>y⁺>Gal4, UAS-GFP^56ST/+; FRT82B tub-Gal80/UAS-Ras^V12 FRT82B scrib²; (C) yw ey-FLP/+; act>y⁺>Gal4, UAS-GFP^56ST/UAS-drICE^RNAi; FRT82B tub-Gal80/UAS-Ras^V12 FRT82B scrib².

DOI: https://doi.org/10.7554/eLife.26747.007

The following figure supplement is available for figure 3:

**Figure supplement 1.** Reduction of ROS results in loss of caspase activity in scrib⁻/⁻ Ras^V12 mosaic discs.

DOI: https://doi.org/10.7554/eLife.26747.008

appearance at neoplastic scrib⁻/⁻ Ras^V12 discs. They single out from the cellular clusters and develop cellular protrusions, similar to cytonemes (*Figure 4M*, arrows). This change in cellular behavior and morphology of hemocytes was also observed in the 'undead' AiP model (*Fogarty et al., 2016*).

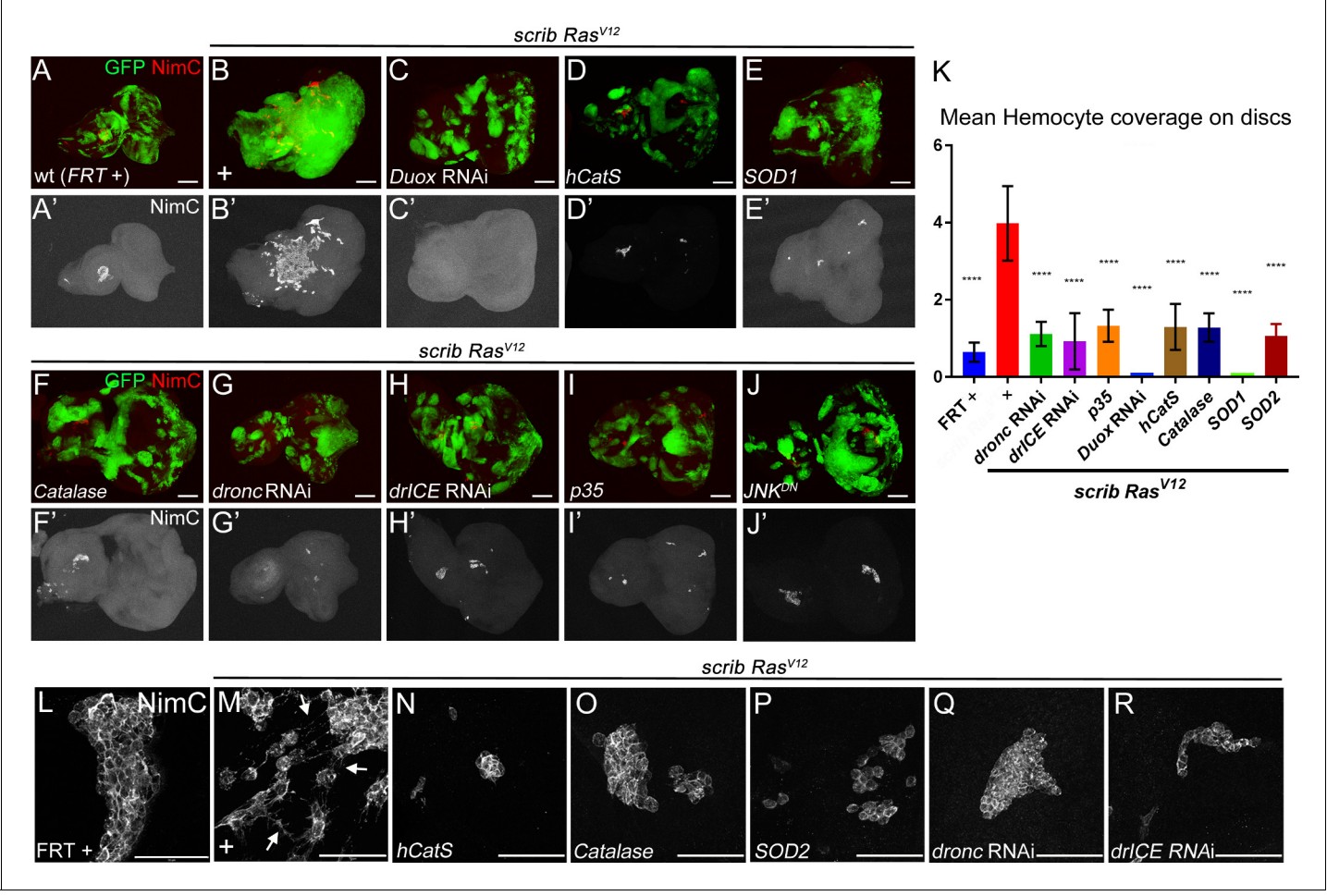

**Figure 4.** Caspase-generated ROS are required of recruitment and activation of hemocytes to *scrib*$^{-/-}$ *Ras*$^{V12}$ mosaic eye/antennal imaginal discs. Hemocytes were labeled with the NimC antibody (*Kurucz et al., 2007*) (red in top panels; grey in (') panels and in (**L–R**)). Scale bars in (**A–J'**): 50 µm; in (**L–R**): 100 µm. (**A,A'**) Control mosaic discs (*FRT* +) carry small hemocyte clusters mostly at the antennal portion of the disc. (**B,B'**) *scrib*$^{-/-}$ *Ras*$^{V12}$ mosaic discs are covered by large quantities of hemocytes. + indicates *scrib*$^{-/-}$ *Ras*$^{V12}$ in otherwise wt background. (**C–I**) Hemocyte recruitment to *scrib*$^{-/-}$ *Ras*$^{V12}$ eye/antennal imaginal discs is strongly impaired upon loss of ROS (**C–F'**) and caspase activity (**G–I'**). (**J,J'**) Expression of a dominant negative JNK transgene (*JNK*$^{DN}$) in *scrib*$^{-/-}$ *Ras*$^{V12}$ mutant clones blocks hemocyte recruitment. (**K**) Quantification of NimC labelings reveals that the number of hemocytes attached to *scrib*$^{-/-}$ *Ras*$^{V12}$ mosaic discs is significantly lower when ROS levels or caspase activity are reduced in *scrib*$^{-/-}$ *Ras*$^{V12}$ clones. To facilitate the quantification, the mean intensity of NimC labelings across the entire disc was determined, normalized to GFP (to account for the reduced size of ROS-depleted or caspase-inhibited *scrib*$^{-/-}$ *Ras*$^{V12}$ clones) and analyzed by one-way ANOVA with Holm-Sidak test for multiple comparisons. Error bars are SD. P values are referenced to *scrib*$^{-/-}$ *Ras*$^{V12}$ and are indicated by asterisks above each column. ****$p<0.0001$. Ten discs per genotype were analyzed. + indicates *scrib*$^{-/-}$ *Ras*$^{V12}$ in otherwise wt background. (**L–R**) High magnification images of hemocytes attached to the discs of indicated genotype. Note that in (**M**) hemocytes attached to *scrib*$^{-/-}$ *Ras*$^{V12}$ discs extend cellular protrusions (arrows), similar to cytonemes. These protrusions are absent in hemocytes attached to control (**L**) and caspase-inhibited or ROS-depleted discs (**N–R**). + in (**M**) indicates *scrib*$^{-/-}$ *Ras*$^{V12}$ in otherwise wt background. **Genotypes**: (**A,L**) *yw ey-FLP/+; act>y*$^+$*>Gal4, UAS-GFP*$^{56ST}$*/+; FRT82B tub-Gal80/. FRT82B w*$^+$; (**B,M**) *yw ey-FLP/+; act>y*$^+$*>Gal4, UAS-GFP*$^{56ST}$*/+; FRT82B tub-Gal80/UAS-Ras*$^{V12}$ *FRT82B scrib*$^2$; (**C–I, N–R**) *yw ey-FLP/+; act>y*$^+$*>Gal4, UAS-GFP*$^{56ST}$*/UAS-X; FRT82B tub-Gal80/UAS-Ras*$^{V12}$ *FRT82B scrib*$^2$ with *UAS-X* being *UAS-Duox*$^{RNAi}$ (**C**), *UAS-hCatS* (**D,N**), *UAS-SOD1* (**E**), *UAS-Catalase* (**F,O**), *UAS-SOD2* (**P**), *UAS-dronc*$^{RNAi}$ (**G,Q**), *UAS-drICE*$^{RNAi}$ (**H,R**) and *UAS-p35* (**I**). (**J**) *yw ey-FLP/UAS-JNK*$^{DN}$*; act>y*$^+$*>Gal4, UAS-GFP*$^{56ST}$*/+; FRT82B tub-Gal80/UAS-Ras*$^{V12}$ *FRT82B scrib*$^2$.

DOI: https://doi.org/10.7554/eLife.26747.009

We examined if ROS and caspases are required for hemocyte recruitment to *scrib*$^{-/-}$ *Ras*$^{V12}$ mosaic discs. Indeed, reduction of ROS strongly reduces the recruitment of hemocytes to *scrib*$^{-/-}$ *Ras*$^{V12}$ mosaic discs (*Figure 4C–F'*). Likewise, the recruitment of hemocytes to *scrib*$^{-/-}$ *Ras*$^{V12}$ tumors is impaired by loss of caspase activity (*Figure 4G–I'*). Quantification of hemocyte recruitment to ROS-depleted or caspase-inhibited *scrib*$^{-/-}$ *Ras*$^{V12}$ discs normalized to GFP (to account for the

reduced size of ROS-depleted or caspase-inhibited $scrib^{-/-}$ $Ras^{V12}$ clones), revealed a significant loss of hemocytes compared to $scrib^{-/-}$ $Ras^{V12}$ mosaic discs (**Figure 4K**). For example, more than 90% of $scrib^{-/-}$ $Ras^{V12}$ discs expressing *Duox* RNAi are not attached by any hemocyte despite the presence of many small clones (**Figure 4C'**). Furthermore, in addition to the significant loss of hemocytes, the few hemocytes that are attached to ROS- and caspase-depleted *scrib* $Ras^{V12}$ discs (**Figure 4D'–I'**), display the naive morphology seen at control discs (**Figure 4N–R**). These observations provide strong evidence that caspase-dependent generation of ROS is essential for recruitment and activation of hemocytes to $scrib^{-/-}$ $Ras^{V12}$ tumors.

## Caspase activation and ROS generation depends on JNK signaling

As reported previously, JNK activity is strongly induced in $scrib^{-/-}$ $Ras^{V12}$ clones (**Figure 5C**) and is essential for the neoplastic phenotype of $scrib^{-/-}$ $Ras^{V12}$ mosaic animals (**Brumby and Richardson, 2003**; **Igaki et al., 2006**; **Igaki et al., 2009**; **Uhlirova et al., 2005**; **Leong et al., 2009**; **Cordero et al., 2010**). In fact, activation of JNK in oncogenic $Ras^{V12}$ background is sufficient to trigger a neoplastic tumor phenotype similar to the $scrib^{-/-}$ $Ras^{V12}$ condition (**Uhlirova et al., 2005**; **Uhlirova and Bohmann, 2006**). Therefore, we examined the relation between ROS, caspases and JNK signaling. In a first set of experiments, we blocked JNK signaling by expressing a dominant negative *JNK* construct ($JNK^{DN}$) in $scrib^{-/-}$ $Ras^{V12}$ mutant clones. In $JNK^{DN}$-expressing $scrib^{-/-}$ $Ras^{V12}$ mutant clones, caspase activity (CC3) and ROS production are strongly reduced (**Figure 5A–B'**). Likewise, the recruitment and activation of hemocytes is strongly impaired at $JNK^{DN}$; $scrib^{-/-}$ $Ras^{V12}$ discs (**Figure 4J**) consistent with a previous report (**Külshammer and Uhlirova, 2013**). These findings suggest that caspase activation, ROS generation and hemocyte activation are dependent on JNK signaling.

In the second set of experiments, we examined if there is a dependence of JNK signaling on caspases and ROS using anti-phosphoJNK (pJNK) antibody as a JNK activity marker. These labelings revealed a significant loss of JNK activity in caspase-inhibited or ROS-depleted $scrib^{-/-}$ $Ras^{V12}$ clones (**Figure 5D–K'**; quantified in **Figure 5L**). Similar results were obtained using MMP1 antibody labeling as an additional JNK marker (**Figure 5—figure supplement 1**). These observations suggest that the maintenance of JNK activity requires caspases and ROS. Combined, these results imply that JNK is acting both upstream (**Figure 5A,B**) and downstream (**Figure 5E–L**) of caspases and ROS. The easiest way to explain such a behavior is that caspases, ROS, hemocytes and JNK signaling constitute an amplification loop in $scrib^{-/-}$ $Ras^{V12}$ mutant clones similar to the 'undead' AiP model (**Diwanji and Bergmann, 2017a**; **Fogarty et al., 2016**).

## Discussion

The traditional view of caspases holds that they counter tumorigenesis by eliminating tumor cells and thus mediate a tumor suppressor function (**Hanahan and Weinberg, 2000**; **Hanahan and Weinberg, 2011**). This tumor-suppressing function of caspases has been reported in mammalian systems (**Asselin-Labat et al., 2011**; **Ho et al., 2009**) and in *Drosophila*, where, for example, *scrib* mutant cells undergo caspase-dependent apoptosis in a JNK- and Eiger-dependent manner (**Brumby and Richardson, 2003**; **Igaki et al., 2006**; **Igaki et al., 2009**; **Uhlirova et al., 2005**). However, more recent work has suggested that caspases and apoptosis in general can also have the opposite, tumor-promoting function, both in flies and in mammals (**Cheng et al., 2015**; **Donato et al., 2014**; **Huang et al., 2011**; **Kurtova et al., 2015**; **Li et al., 2010**; **Zhang et al., 2015**); reviewed by (**Ryoo and Bergmann, 2012**; **Ichim and Tait, 2016**; **Fogarty and Bergmann, 2017**). Furthermore, in *Drosophila* it was previously shown that oncogenic Ras switches the tumor-suppressing function of JNK and Eiger in $scrib^{-/-}$ mutant cells to a tumor-promoting one in $scrib^{-/-}$ $Ras^{V12}$ cells (**Igaki et al., 2006**; **Cordero et al., 2010**; **Uhlirova et al., 2005**; **Enomoto et al., 2015**). Mechanistic details about this oncogenic switch have been largely elusive. Our data presented here imply that a critical step for this oncogenic switch is the conversion of caspase activity by oncogenic Ras.

Consistent with a previous report in different context (**Hirabayashi et al., 2013**), our results demonstrated that oncogenic Ras can suppress the apoptotic activity of caspases and keeps $scrib^{-/-}$ $Ras^{V12}$ cells in an 'undead'-like condition. While caspases may still induce apoptosis in a few tumor cells, they now largely promote generation of intra- and extracellular ROS which are required for malignant growth and tissue invasion of surviving neoplastic cells. Evidence that caspases are indeed

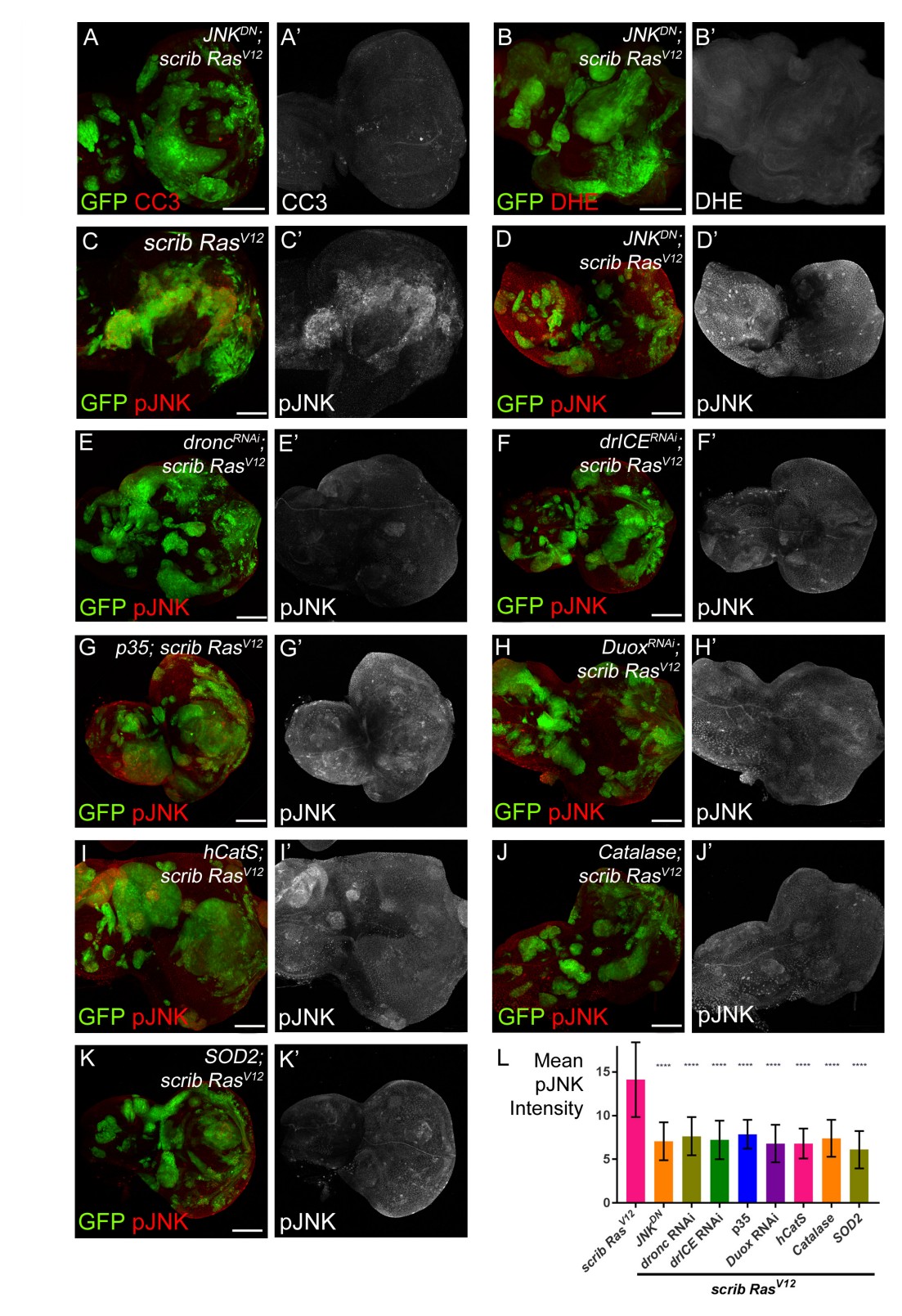

**Figure 5.** JNK acts upstream and downstream of caspase activation and ROS generation. (A–B') Expression of *JNK^DN* in *scrib^−/−^ Ras^V12^* clones inhibits caspase activity (A,A'; CC3) and ROS generation (B,B'; DHE). Scale bars: 50 μm. (C–K') pJNK labeling (red in (C–K); grey in (C'–K')) as JNK marker in *scrib^−/−^ Ras^V12^* (C,C'), *JNK^DN^*-expressing *scrib^−/−^ Ras^V12^* (D,D') and ROS-depleted or caspase-inhibited *scrib^−/−^ Ras^V12^* mosaic discs (E–K'). The strong pJNK labeling in (C,C') is significantly reduced in (D–K'). Scale bars: 50 μm. (L) The mean intensity of pJNK labelings in *scrib^−/−^ Ras^V12^* clones in panels

*Figure 5 continued on next page*

*Figure 5 continued*

(C'–K') is significantly reduced upon ROS-depletion or reduction of caspase activity. Analysis of JNK labelings was done by one-way ANOVA with Holm-Sidak test for multiple comparisons. Error bars are SD. P values are referenced to $scrib^{-/-}$ $Ras^{V12}$ and are indicated by asterisks above each column. ****p<0.0001. At least ten discs per genotype were analyzed. Genotypes: (A,B,D) *yw ey-FLP/UAS-JNK*[DN]; *act>y*[+]*>Gal4, UAS-GFP*[56ST]/+; *FRT82B tub-Gal80/UAS-Ras*[V12] *FRT82B scrib*[2]; (C) *yw ey-FLP/+; act>y*[+]*>Gal4, UAS-GFP*[56ST]/+; *FRT82B tub-Gal80/UAS-Ras*[V12] *FRT82B scrib*[2]; (E–K) *yw ey-FLP/+; act>y*[+]*>Gal4, UAS-GFP*[56ST]/*UAS-X*; *FRT82B tub-Gal80/UAS-Ras*[V12] *FRT82B scrib*[2] with *UAS-X* being *UAS-dronc*[RNAi] (E), *UAS-drICE*[RNAi] (F), *UAS-p35* (G), *UAS-Duox*[RNAi] (H), *UAS-hCatS* (I), *UAS-Catalase* (J) and *UAS-SOD2* (K).
DOI: https://doi.org/10.7554/eLife.26747.010
The following figure supplement is available for figure 5:

**Figure supplement 1.** MMP1 labeling is reduced in $scrib^{-/-}$ $Ras^{V12}$ clones with reduced ROS or caspase activity.
DOI: https://doi.org/10.7554/eLife.26747.011

activated in $scrib^{-/-}$ $Ras^{V12}$ clones is not only provided in this study, but also in a report that showed that small malignant clones ($lgl^{-/-}$ $Ras^{V12}$) undergo caspase-mediated apoptosis and elimination in a similar way as *scrib* clones (**Menéndez et al., 2010**). Only when mutant clones have reached a certain size, can they develop malignant tumors despite intrinsic caspase activation (**Menéndez et al., 2010**; **Ballesteros-Arias et al., 2014**).

Our data suggest that JNK activity acts both upstream and downstream of caspases and ROS generation in $scrib^{-/-}$ $Ras^{V12}$ mutant clones (**Figure 5**). It is possible that initially, when the $scrib^{-/-}$ $Ras^{V12}$ mutant cells form, a cell competition signal triggers JNK activation in $scrib^{-/-}$ $Ras^{V12}$ cells, similar to the events in $scrib^{-/-}$-only mutant cells (**Figure 6**). In both $scrib^{-/-}$ and $scrib^{-/-}$ $Ras^{V12}$ mutant cells, this JNK activity results in caspase activation (**Figures 3** and **5A**). However, due to the anti-apoptotic activity of $Ras^{V12}$, caspases induce only very little apoptosis in *scrib* $Ras^{V12}$ clones (**Figure 3**). Therefore, although caspases are active, most $scrib^{-/-}$ $Ras^{V12}$ cells do not die and thus are in an 'undead'-like state. Caspases now promote the generation of extracellular ROS through activation of NADPH oxidase Duox. These ROS recruit and activate hemocytes (**Figure 4**). It is known that hemocytes can release Eiger (**Cordero et al., 2010**; **Parisi et al., 2014**; **Fogarty et al., 2016**) which signals through its receptor Grindelwald (**Andersen et al., 2015**) for further stimulation of JNK activity in $scrib^{-/-}$ $Ras^{V12}$ cells. Thus, we postulate that JNK, caspases, ROS, hemocytes and Eiger constitute an amplification loop (**Figure 6**) which may be necessary for tumor initiation.

Evidence of an amplification loop is provided by the mutual dependence of caspases, ROS and JNK (**Figure 5**; **Figure 3—figure supplement 1**). Similar amplification loops have been described in apoptotic and 'undead' cells (**Wells et al., 2006**; **Shlevkov and Morata, 2012**; **Fogarty et al., 2016**). This amplification loop ensures persistent oncogenic signaling in $scrib^{-/-}$ $Ras^{V12}$ cells. This is in striking contrast to $scrib^{-/-}$ mutant cells alone in which JNK signaling triggers linear caspase activation and apoptosis (**Brumby and Richardson, 2003**; **Igaki et al., 2006**; **Igaki et al., 2009**; **Uhlirova et al., 2005**). In later stages of tumorigenesis, the amplification loop may have reached full strength and promotes malignant growth and invasion of $scrib^{-/-}$ $Ras^{V12}$ cells (**Figure 6**). Amplification loops have also been observed in other neoplastic tumor models in *Drosophila*. For example, intestinal stem cell tumors form in response to an amplification loop (**Chen et al., 2016**). In a glycolytic tumor model, ROS are also part of a amplification loop that facilitates metabolic reprogramming (**Wang et al., 2016**). Thus, it is possible that tumorigenesis in general depends on amplification loops to sustain oncogenic signaling.

There are similarities and differences in the amplification loops of the 'undead' hyperplastic AiP model and the neoplastic $scrib^{-/-}$ $Ras^{V12}$ tumor model. In both models, caspases, ROS, hemocytes and JNK are required for growth. However, regarding caspases, the 'undead' AiP model only involves the initiator caspase Dronc for growth (in fact, effector caspases are inhibited by P35 in this model) (**Fan et al., 2014**; **Huh et al., 2004**; **Pérez-Garijo et al., 2004**; **Pérez-Garijo et al., 2009**; **Ryoo et al., 2004**). In contrast, the neoplastic $scrib^{-/-}$ $Ras^{V12}$ model requires both initiator (Dronc) and effector (DrICE) caspases. Inhibition of either suppresses malignant growth and invasion. Another interesting difference is the differential involvement of ROS. While only extracellular ROS are essential in the 'undead' AiP model, neoplastic growth of $scrib^{-/-}$ $Ras^{V12}$ cells requires both intra- and extracellular ROS. Mitochondria are the likely source of intracellular ROS because expression of mitochondrial SOD2 can suppress tumor growth and invasion of $scrib^{-/-}$ $Ras^{V12}$ tumors (**Figure 1N**). It is unclear if these two populations of ROS are dependent or independent of each

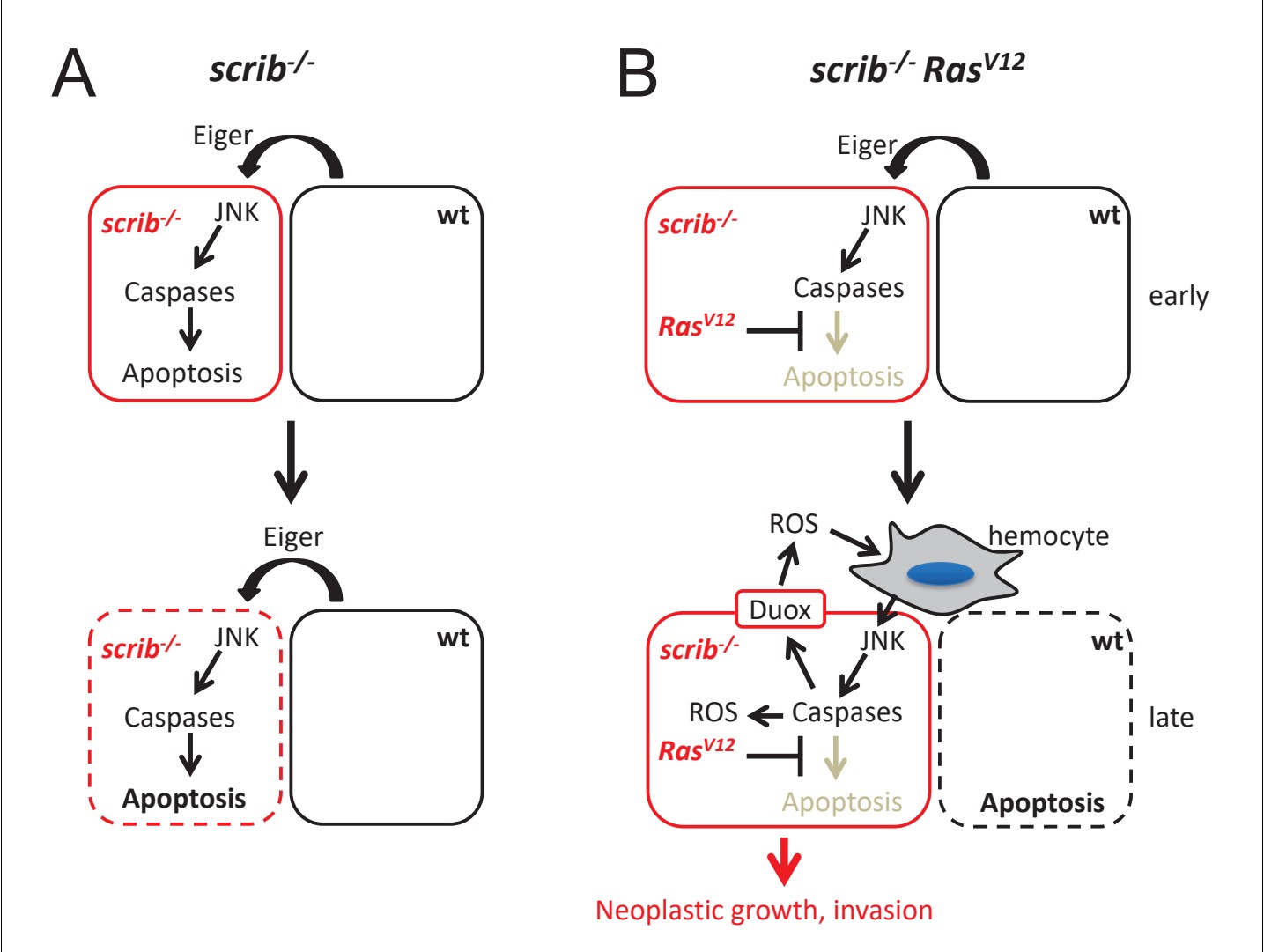

**Figure 6.** Mechanistic view about the conversion of caspases from tumor suppressors to tumors promoters in *scrib Ras*$^{V12}$ mutant cells. (**A**) After *scrib*$^{-/-}$ mutant cells have formed in mosaic discs, a cell competition signal mediated by Eiger triggers JNK and caspase activity which induces apoptosis of *scrib* mutant cells (dashed outline of the cell). (**B**) After *scrib*$^{-/-}$ *Ras*$^{V12}$ cells have formed, JNK activity may be induced by the same cell competition signal as in *scrib*$^{-/-}$ mutant cells (early). However, despite activation of JNK and caspases in *scrib Ras*$^{V12}$ cells, Ras$^{V12}$ keeps these cells in an 'undead'-like condition and enables caspases to initiate a feedback amplification loop involving ROS generation and recruitment of hemocytes which amplifies JNK and caspase activity (late). This amplification loop is necessary for malignant growth and invasion. Neighboring wild-type cells undergo apoptosis (dashed outline of the cell). The mechanism of non-autonomous apoptosis is not clear.

DOI: https://doi.org/10.7554/eLife.26747.012

other. Recently, it was shown in endothelial cells that Nox-derived ROS can trigger an increase in mitochondrially-derived ROS and loss of mitochondrial membrane potential suggesting a dependence of mitochondrial ROS from Nox-generated ROS (*Shafique et al., 2017*). However, alternatively, it is also possible that intra- and extracellular ROS are produced independently by caspase activity, and both are separately required for neoplastic transformation.

Furthermore, while we mostly focused from the point-of-view of 'undead' AiP, it is also possible that 'genuine' AiP contributes to the tumor growth in *scrib*$^{-/-}$ *Ras*$^{V12}$. Consistent with this notion is the observation that there are a few apoptotic cells in *scrib*$^{-/-}$ *Ras*$^{V12}$ mutant clones (*Figure 3E',F*). Genuinely apoptotic cells can also produce ROS in *Drosophila* imaginal discs (*Santabárbara-Ruiz et al., 2015*). Therefore, it is tempting to speculate that some of the differences between the 'undead' AiP model and the *scrib*$^{-/-}$ *Ras*$^{V12}$ model are due to a combination of 'undead' and

'genuine' AiP in $scrib^{-/-}\ Ras^{V12}$. This may explain the differences in caspase requirement, the differences in ROS production and the different outcomes of growth – hyperplastic vs. neoplastic – between the 'undead' AiP and the $scrib^{-/-}\ Ras^{V12}$ tumor models. Future work will address this important question.

Another important question for future studies will be to address how $Ras^{V12}$ switches the activity of caspases from tumor-suppressors to tumor-promoters. A known target of survival signaling by $Ras^{V12}$ is the pro-apoptotic gene Hid which acts upstream of caspase activation (*Bergmann et al., 1998*; *Kurada and White, 1998*). Because Eiger and JNK can induce expression of *hid* (*Moreno et al., 2002*), it is possible that Hid activity is inhibited by oncogenic Ras. However, while we do not exclude this possibility, it alone may not be sufficient to explain the altered caspase activity because inhibition of *hid* would result in loss of caspase activity. However, caspases are still active in $scrib^{-/-}\ Ras^{V12}$ mutant cells (*Figure 3B,D*) and they are also able to induce apoptosis at least in a small amount of mutant cells (*Figure 3E,F*). Therefore, it is possible that $Ras^{V12}$ modifies caspase activity in a different manner – directly or indirectly – for non-apoptotic ROS generation.

Oncogenic Ras is mediating many steps in the tumorigenic process of $scrib^{-/-}\ Ras^{V12}$ tissue. It changes the transcriptome of these cells and modifies the downstream activities of JNK (*Atkins et al., 2016*; *Külshammer et al., 2015*). However, as shown in this work, a critical step mediated by $Ras^{V12}$ is the modification of caspase activity – directly or indirectly – in the early stage of tumorigenesis in a way that the cells survive. At present it is unclear if the caspase-modulating activity of $Ras^{V12}$ is dependent on transcription. There may not be enough time for a transcriptional response by the cell to escape the apoptotic activity of caspases. Consistently, work by others has suggested that changes in the transcriptome alone does not fully explain the neoplastic phenotype of $scrib^{-/-}\ Ras^{V12}$ cells and that other potentially non-transcriptional processes are involved (*Atkins et al., 2016*). Modification of caspase activity may be one of these non-transcriptional processes.

Non-apoptotic functions of caspases have been reported (*Shalini et al., 2015*; *Fogarty and Bergmann, 2017*; *Mukherjee and Williams, 2017*; *Nakajima and Kuranaga, 2017*). However, it is largely unknown how cells survive in the presence of activated caspases during non-apoptotic processes. It is possible that a reduction of caspase activity below a certain apoptotic threshold is sufficient for survival. Other models include changes in the subcellular localization of caspases or interaction with modifying factors such as Tango7 (*D'Brot et al., 2013*). Interesting in this respect is a recent study which showed that mitochondrially-derived SCSβ restricts caspase activity for spermatid maturation in the *Drosophila* testis (*Aram et al., 2016*). Since mitochondria release apoptotic signaling molecules such as cytochrome c and Smac during apoptosis (*Fuchs and Steller, 2011*), it may also be possible that they release signals such as SCSβ or related factors which modulate the activity of caspases for non-apoptotic functions. More work is necessary to address these essential questions for understanding of tumor initiation and progression.

## Materials and methods

### *Drosophila* genetics

The *scrib* allele used is $scrib^2$ (also known as $scrib^{673}$) (*Bilder and Perrimon, 2000*). The recombinant $UAS\text{-}Ras^{V12}\ FRT82B\ scrib^2$ (*Chen et al., 2012*; *Pérez et al., 2015*) line was a kind gift of Madhuri Kango-Singh (U Dayton, OH, USA). The MARCM system (*Lee and Luo, 1999*) with *ey-FLP* (*Newsome et al., 2000*) was used to generate mosaics of eye/antennal imaginal discs and experimental clones were marked by GFP. In (*Figure 1—figure supplement 1*), we used a modified MARCM system that marks clones with myrRFP (*Chabu and Xu, 2014*). The wt control line (*FRT +*) is *FRT82B* (*Xu and Rubin, 1993*).

The following transgenes are all inserted on chromosome 2 and were crossed into the $scrib^{-/-}\ Ras^{V12}$ background for analysis: *UAS-lacZ* (Bloomington, BL3955), *UAS-Duox* RNAi and *UAS-hCatS* (a kind gift of Won-Jae Lee) (*Ha et al., 2005a*; *Ha et al., 2005b2005*), *UAS-Catalase* (BL24621), *UAS-SOD1* (BL24754), *UAS-SOD2* (BL24494), *UAS-p35* (BL5072), *UAS-dronc* RNAi and *UAS-drICE RNAi* (a kind gift of Pascal Meier) (*Leulier et al., 2006*). $UAS\text{-}JNK^{DN}$ (aka $UAS\text{-}bsk^{DN}$) (BL6409) is an insertion on X chromosome. Crosses were incubated at either 22° or 25°C.

## Imaging and quantification

DHE and $H_2DCF-DA$ (both from Invitrogen/Molecular Probes) labeling of unfixed tissue was performed as described (*Owusu-Ansah et al., 2008*). TUNEL labeling (Roche) was done according to the manufacturer's instructions. Antibody labelings were done on fixed tissue following standard procedures (*Fan and Bergmann, 2014*; *Fogarty and Bergmann, 2014*). The following antibodies were used: cleaved caspase-3 (CC3; Cell Signaling Technology); NimC (kind gift of I. Andó) (*Kurucz et al., 2007*); MMP1 (Developmental Studies Hybridoma Bank (DSHB)) and pJNK (Promega). Secondary antibodies were donkey Fab fragments from Jackson Immunoresearch. Eye/antennal cephalic complexes were counterlabeled with the nuclear dye DAPI to visualize tissue outline. Images were taken with a Zeiss LSM700 confocal microscope. For quantification of confocal images, the 'Record Measurement' function of Photoshop was used. Clones were outlined and signal intensity determined. Multiple clones of five to ten imaginal discs per genotype obtained in three independent experiments were measured. Analysis and graph generation was done using GraphPad Prism 7.03. The statistical method and the P values are indicated in the figure legends.

## Acknowledgements

We would like to thank István Andó, Madhuri Kango-Singh, Won Jae Lee, Pascal Meier, Tian Xu, the Bloomington and Vienna stock centers for fly stocks and reagents; Eric Baehrecke, Yun Fan and Caitlin E. Fogarty for stimulating discussions. This work was supported by MIRA award R35 GM118330 from the NIH/NIGMS. The content is solely the responsibility of the authors and does not necessarily represent the official views of the National Institutes of Health.

## Additional information

### Funding

| Funder | Grant reference number | Author |
| --- | --- | --- |
| National Institute of General Medical Sciences | R35GM118330 | Andreas Bergmann |
| National Institute of General Medical Sciences | R01GM107789 | Andreas Bergmann |

The funders had no role in study design, data collection and interpretation, or the decision to submit the work for publication.

### Author contributions

Ernesto Pérez, Conceptualization, Formal analysis, Supervision, Funding acquisition, Writing—original draft, Writing—review and editing; Jillian L Lindblad, Conceptualization, Data curation, Formal analysis, Investigation, Writing—original draft; Andreas Bergmann, Data curation, Formal analysis, Investigation

### Author ORCIDs

Andreas Bergmann iD http://orcid.org/0000-0002-9134-871X

### Decision letter and Author response

Decision letter https://doi.org/10.7554/eLife.26747.014
Author response https://doi.org/10.7554/eLife.26747.015

## Additional files

### Supplementary files

• Transparent reporting form
DOI: https://doi.org/10.7554/eLife.26747.013

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
