## [Decision Letter]

Thank you for submitting your article "Tumor-promoting function of apoptotic caspases by an amplification loop involving ROS, macrophages and *JNK* in *Drosophila*" for consideration by *eLife*. Your article has been reviewed by three peer reviewers, one of whom is a member of our Board of Reviewing Editors, and the evaluation has been overseen by K VijayRaghavan as the Senior Editor. The following individual involved in review of your submission has agreed to reveal his identity: Pascal Meier

The reviewers have discussed the reviews with one another and the Reviewing Editor has drafted this decision to help you prepare a revised submission.

Summary:

While caspases and cell death are generally thought to play tumour suppressive roles, more and more evidence from *Drosophila* and mammals indicate that caspases in fact can have tumour promoting roles. Here, Bergmann and co-workers use a *Drosophila* Scribble (Scrib)/*Ras*-^V12^ tumour model to assess the contribution of caspases. They find that the effector caspases contribute to tumour growth. In particular, they find that active *drICE* contributes to the production of intra- and extra-cellular ROS. The produced ROS in turn attracts hemocytes (*Drosophila* immune cells), which drive Eiger-dependent and *JNK*-mediated tumour growth.

Essential revisions:

The authors must quantify the data in the manuscript (as detailed below), as well as provide additional background information, and soften some conclusions or provide more support for these statements.

1) Several points are vague and often lacks needed quantification. For example when examining DHE staining of *scrib-/- ras^V12^* clones, they comment that some clones have diffuse cytosolic staining, others stain intensely, while others have web-like staining. No attempt is made to define these differences. The DHE staining must be quantified. Other readouts of ROS could be useful. Similarly, when examining ROS and *JNK*, in *scrib-/ ras^V12^* clones the authors state that "many clones" that were depleted of ROS or had inhibited caspase had less *JNK*. How many and what percent of clones lose *JNK*? How many (and what percent) of ey>MARCM *scrib-/-; ras^v12^* adult animals live when ROS is inhibited ?

Quantification of hemocyte numbers/area they cover relative to the total area of the clones is also needed for Figure 4. The number of hemocytes seems very different from Figure 4 to Figure 4

2) The authors observe Cc3 and TUNEL staining inside and outside of *scrib-/- ras^v12^* clones. The authors suggest that this death is essential to the tumor, and that the clones are supercompetitors. If they want to support this claim, they should inhibit death only in the surrounding wildtype cells and see if this is essential. Alternatively they can modify this in the Discussion.

3) Introduction: At present, the link to mammalian settings is insufficient, and the manuscript would benefit from more clearly spelling out previous observations hinting at a pro-tumorigenic role of caspases. Given the numerous studies that demonstrate tumor promoting roles of caspases, it might be appropriate to better reflect the current state of knowledge on this issue.

4)The authors suggest that cell death of non-*Scrib/Ras^V12^* cells contribute to the growth of *Scrib/Ras^V12^* tumor cells. It would be interesting to test this hypothesis, and evaluate the contribution of *JNK* activation and cell death of neighbouring wild-type cells (inactivating caspases only in wild-type cells). Although this would be interesting to address, I realize that by-stander cell death is not exactly to the issue of this manuscript. Hence, this could be an added benefit, but is non-essential for the current study.

5) Figure 3 would profit from a side-by-side comparison with *Scrib* and *Scrib/Ras^V12^/drICE* RNAi. While the results from the TUNEL staining are clear and convincing, it is somewhat difficult to evaluate the extent of caspase activation in Figure 3. Clearly, inhibition of caspase activation suppresses ROS production, hemocyte recruitment and tumour overgrowth. The phenotype is clear. But it is difficult to evaluate the extent of caspase activation in the *Scrib/Ras^V12^* clones.

It is important to show double staining of CC3 (cleaved caspase3) and TUNEL in the *scrib Ras^V12^* clones. CC3 staining outside the clones, as shown in Figure 3, looks different to the TUNEL staining in Figure 3. Furthermore, staining with CC3 and TUNEL would be the way to show which cells are indeed activating caspases without undergoing cell death. CC3 staining should be performed in *scrib Ras^V12^* Duoxi, hCatS, SOD1, SOD2, Catalase… to provide a stronger evidence for the loop. It would also be nice to have CC3 staining in p35, dronci and dricei as controls.

6) A very interesting observation is the fact that *JNK* is activated in the periphery of the *scrib Ras^V12^* clones when ROS or caspases are inhibited. Unfortunately, only few not very clear examples are provided, and instead most of the times staining with MMP1 is shown, where this phenomenon can't be observed. pJNK staining (which would be a better readout for *JNK* staining) or alternatively puc-lacZ, should be shown for the rest of the genotypes, including the *scrib Ras^V12^* alone, which is not shown in the paper. Again, it would also be nice to have this staining in the JNKDN experiment as a control.

---

## [Author Response]

Essential revisions:The authors must quantify the data in the manuscript (as detailed below), as well as provide additional background information, and soften some conclusions or provide more support for these statements.

As requested, we have quantified all data in the manuscript. We have provided additional background information in the Introduction and have softened controversial statements.

1) Several points are vague and often lacks needed quantification. For example when examining DHE staining of scrib-/- ras^V12^ clones, they comment that some clones have diffuse cytosolic staining, others stain intensely, while others have web-like staining. No attempt is made to define these differences. The DHE staining must be quantified. Other readouts of ROS could be useful. Similarly, when examining ROS and JNK, in scrib-/ ras^V12^ clones the authors state that "many clones" that were depleted of ROS or had inhibited caspase had less JNK. How many and what percent of clones lose JNK? How many (and what percent) of ey>MARCM scrib-/-; ras^v12^ adult animals live when ROS is inhibited ?Quantification of hemocyte numbers/area they cover relative to the total area of the clones is also needed for Figure 4. The number of hemocytes seems very different from Figure 4 to Figure 4

We agree with the reviewers that quantification is needed to support our conclusions. Therefore, we have spent a considerable amount of time to quantify all the data. This includes the DHE, pJNK, MMP1, hemocytes, cleaved caspase-3 and TUNEL labelings. We have also added and quantified H_2_DCF-DA labelings as additional ROS readout in Figure 1—figure supplement 1. The H_2_DCF-DA data are very consistent with the DHE labelings. We have also significantly expanded on the counts of pupal and adult survival and have now much more reliable numbers for all genotypes (Figure 1 and Figure 2). Overall, the quantified data are very consistent with our conclusions.

Figure 4 are not concerned with the number of hemocytes, but rather with the morphology of hemocytes on *scrib^-/-^ Ras^V12^* discs and discs that have reduced caspase activity and reduced ROS. We have clarified this point in the text.

2) The authors observe Cc3 and TUNEL staining inside and outside of scrib-/- ras^v12^ clones. The authors suggest that this death is essential to the tumor, and that the clones are supercompetitors. If they want to support this claim, they should inhibit death only in the surrounding wildtype cells and see if this is essential. Alternatively they can modify this in the Discussion.

In the long term, it is certainly a very important question to address if the apoptotic cell death outside the clones (non-autonomously) is essential for tumor growth. However, experimentally, it is not possible to address the allotted time of the revision. We have therefore removed these statements and any reference to supercompetitors.

3) Introduction: At present, the link to mammalian settings is insufficient, and the manuscript would benefit from more clearly spelling out previous observations hinting at a pro-tumorigenic role of caspases. Given the numerous studies that demonstrate tumor promoting roles of caspases, it might be appropriate to better reflect the current state of knowledge on this issue.

We agree with the reviewers and have expanded and clarified the different forms of AiP and the role of caspases for tumor growth in mammals. We also have cited all relevant original papers on this topic.

4)The authors suggest that cell death of non-Scrib/Ras^V12^ cells contribute to the growth of Scrib/Ras^V12^ tumor cells. It would be interesting to test this hypothesis, and evaluate the contribution of JNK activation and cell death of neighbouring wild-type cells (inactivating caspases only in wild-type cells). Although this would be interesting to address, I realize that by-stander cell death is not exactly to the issue of this manuscript. Hence, this could be an added benefit, but is non-essential for the current study.

This point is similar to point #2. To inactivate caspases only in wild-type cells in a *scrib^-/-^ Ras^V12^* mosaic background is experimentally quite challenging and not possible in the allocated time of the revision.

5) Figure 3 would profit from a side-by-side comparison with Scrib and Scrib/Ras^V12^/drICE RNAi. While the results from the TUNEL staining are clear and convincing, it is somewhat difficult to evaluate the extent of caspase activation in Figure 3. Clearly, inhibition of caspase activation suppresses ROS production, hemocyte recruitment and tumour overgrowth. The phenotype is clear. But it is difficult to evaluate the extent of caspase activation in the Scrib/Ras^V12^ clones.

We have added the *scrib^-/-^ Ras^V12^ /drICE* RNAi analysis to Figure 3. We have omitted a *scrib* analysis, because *scrib* clones are very small. In order to address the extent of caspase activation in *scrib^-/-^ Ras^V12^* clones, we have quantified the CC3 data inside and outside the clones (autonomously and non-autonomously) in control (wild-type mosaics), *scrib^-/-^ Ras^V12^* and *scrib^-/-^ Ras^V12^ /drICE* RNAi mosaics. We found that the caspase activity outside *scrib^-/-^ Ras^V12^* clones is about 2.5 times higher than inside the clones. These data are presented in Figure 3.

It is important to show double staining of CC3 (cleaved caspase3) and TUNEL in the scrib Ras^V12^ clones. CC3 staining outside the clones, as shown in Figure 3, looks different to the TUNEL staining in Figure 3. Furthermore, staining with CC3 and TUNEL would be the way to show which cells are indeed activating caspases without undergoing cell death. CC3 staining should be performed in scrib Ras^V12^ Duoxi, hCatS, SOD1, SOD2, Catalase… to provide a stronger evidence for the loop. It would also be nice to have CC3 staining in p35, dronci and dricei as controls.

We performed a large amount of CC3/TUNEL double labelings, but the CC3 signal in these double labelings was too weak to give meaningful and satisfying results. However, the TUNEL labelling in these experiments went very well which we were able to quantify. We found that almost 90% of all TUNEL-positive cells are outside (non-autonomous) of the *scrib^-/-^ Ras^V12^* clones. The remaining TUNEL-positive cells are located inside these clones. Thus, although our CC3/TUNEL double labelings were inconclusive, these statistical data strongly confirm that most apoptotic cells are located outside of *scrib^-/-^ Ras^V12^* cells.

The CC3 and TUNEL labelings outside the clones look different, because one stain is cytosolic (CC3), while the other one is nuclear (TUNEL). We have also added CC3 labelings of *scrib^-/-^ Ras^V12^* with Duox RNAi, hCatS, Catalase and SOD2 in the supplement to Figure 3. In all cases, the CC3 signal is strongly reduced, further strengthening the positive feedback amplification loop.

6) A very interesting observation is the fact that JNK is activated in the periphery of the scrib Ras^V12^ clones when ROS or caspases are inhibited. Unfortunately, only few not very clear examples are provided, and instead most of the times staining with MMP1 is shown, where this phenomenon can't be observed. pJNK staining (which would be a better readout for JNK staining) or alternatively puc-lacZ, should be shown for the rest of the genotypes, including the scrib Ras^V12^ alone, which is not shown in the paper. Again, it would also be nice to have this staining in the JNKDN experiment as a control.

We have replaced the MMP1 labelings in Figure 5 with pJNK stainings and moved the MMP1 labelings as supporting information into the supplement. As requested, we have also added the JNKDN experiment as control. In addition, all data are quantified.